# Effects of ionic strength on gating and permeation of TREK-2 K2P channels

Linus J. Conrad [1,2¤], Peter Proks[1,3], Stephen J. Tucker [1,2,3]*

1 Clarendon Laboratory, Department of Physics, University of Oxford, Oxford, United Kingdom, 2 OXION Initiative in Ion Channels and Disease, University of Oxford, Oxford, United Kingdom, 3 Kavli Institute for Nanoscience Discovery, University of Oxford, Oxford, United Kingdom

¤ Current address: Department of Biomedical Science, University of Sheffield, Sheffield, United Kingdom
* stephen.tucker@physics.ox.ac.uk

## Abstract

In addition to the classical voltage-dependent behavior mediated by the voltage-sensing-domains (VSD) of ion channels, a growing number of voltage-dependent gating behaviors are being described in channels that lack canonical VSDs. A common thread in their mechanism of action is the contribution of the permeating ion to this voltage sensing process. The polymodal K2P K$^+$ channel, TREK2 responds to membrane voltage through a gating process mediated by the interaction of K$^+$ with its selectivity filter. Recently, we found that this action can be modulated by small molecule agonists (e.g. BL1249) which appear to have an electrostatic influence on K$^+$ binding within the inner cavity and produce an increase in the single-channel conductance of TREK-2 channels. Here, we directly probed this K$^+$-dependent gating process by recording both macroscopic and single-channel currents of TREK-2 in the presence of high concentrations of internal K$^+$. Surprisingly we found TREK-2 is inhibited by high internal K$^+$ concentrations and that this is mediated by the concomitant increase in ionic-strength. However, we were still able to determine that the increase in single channel conductance in the presence of BL1249 was blunted in high ionic-strength, whilst its activatory effect (on channel open probability) persisted. These effects are consistent with an electrostatic mechanism of action of negatively charged activators such as BL1249 on permeation, but also suggest that their influence on channel gating is complex.

## Introduction

K2P-channels are involved in diverse physiological processes such as the perception of pain, mood [1, 2] and many other signaling pathways [3]. They are often described as 'background' or 'leak' channels, however, many members of the family display prominent voltage-dependent behavior [4–6] and their activity can be regulated by a diverse array of physical and chemical stimuli [3].

K2P channels do not contain a classical voltage-sensing-domain (VSD) and most, including the subfamily of mechanically-gated K2Ps (TREK/TRAAK), lack an internal bundle-crossing gate making the selectivity filter (SF) the principal gate in these channels [7, 8]. Recently a

**Data Availability Statement:** All relevant data are within the manuscript and its Supporting information files.

**Funding:** This work was supported by the Wellcome Trust (Grant number: 109114/Z/15/Z)

and the Biotechnology and Biological Sciences Research Council (Grant numbers: BB/N009274/1 and BB/S008608/1). The funders had no role in study design, data collection and analysis, decision to publish, or preparation of the manuscript.

**Competing interests:** The authors declare that no competing interests exist.

lower gate comprising a unique arrangement of the inner M4 helices (X-gate) was found in the structure of TASK-1 [9] and an inner gate has been observed in the structure of TASK-2 [10].

The voltage-dependent gating of TREK channels and other K2Ps was described not long after their discovery and cloning [4, 5, 11], but it was not until recently that a more comprehensive analysis of this mechanism was made [6]. Net gating charge movements of ~2.2 $e_0$ were observed and proposed to result from ion movements into the S4 $K^+$ binding site of the selectivity filter during activation. Due to the dependence on the direction of ion movement (with outward driving forces enabling activation) the mechanism was referred to as "flux gating" with other permeant ions such as $Rb^+$ and $Cs^+$ also having a uniquely stabilizing effect [6]. Interestingly, a recent study found structural evidence for a variety of different occupancy patterns in the TREK-1 selectivity filter that also appear related to the stability of the conductive conformation of the filter gate [12].

The intracellular vestibule of $K^+$ channels is of particular importance for their function. The electrostatic environment of the vestibule is shaped by surface charges that favor occupancy with cations [13]. Such surface charges have been shown to have a profound influence on single-channel conductance [14] with the large-conductance BK $K^+$ channels thought to have evolved additional negative charges at the entrance of this cavity to resupply ions to the filter [15].

The effect of ion concentration on channel gating is a hallmark of filter-related gating phenomena such as C-type inactivation [16]. Therefore, in the context of a filter-gated channel and 'flux gating', the resupply and occupancy of S4 with $K^+$ is expected to affect both single channel conductance and stability of the open filter gate itself (i.e. channel open probability, $P_o$).

We recently found evidence for such a phenomenon in the mechanism of action of 'negatively charged activators' such as the TREK channel agonist, BL1249. This negatively charged activator is proposed to bind in a position that exposes a partially charged tetrazole moiety to the permeation pathway in the vicinity of a $K^+$ ion just below the entrance to the filter [17]. Alongside its principal effect on $P_o$, we also observed an increase in single-channel conductance ($\gamma$). Consistent with the evidence from its likely binding site and further macroscopic electrophysiological data, this increase in $\gamma$ was interpreted as the result of an electrostatic effect on ion permeation through the filter.

Effects of agonists on $\gamma$ are rare, though have been observed before for the effect of ivermectin on P2X receptors [18], various TRPV1 agonists [19] and also diazepam on GABA receptors [20], and in most cases such increases often indicate a unique mechanism of action of the drug on channel behavior. However, extreme care must also be taken in the interpretation of such increases because measured changes in open-channel current do not necessarily reflect a real change in the absolute measure of conduction through the channel. In the context of agonists, their stabilization of very short duration openings can reduce the number of opening events that are distorted by filtering thereby resulting in an apparent increase in single channel amplitude [21].

In all previous cases where we have recorded TREK-2 single-channels, the amplitude histograms exhibited skewed distributions, especially in the inward direction [17, 22, 23]. This distortion is likely produced either by very short openings that are not fully resolved, or by a smaller subconductance state, or a combination thereof. Either way, such skewed distributions are often indicative of an unstable open filter gate conformation undergoing rapid structural fluctuations that exceed the temporal resolution of the single-channel recordings.

A kinetic analysis of single-channel recordings would normally serve to determine the origin of this observed increase in conductance. However, wild-type TREK-2 has an intrinsically low $P_o$, and also expresses with high membrane density in most heterologous expression

systems thus making genuine single-channel recordings extremely challenging. We therefore sought to exploit changes in the concentration of internal $K^+$ to probe these mechanisms and to explain the changes in γ previously observed with BL1249. Surprisingly, we found that TREK channels were inhibited in the presence of high internal $K^+$ concentrations $[K^+_{int}]$ and we show that this is due to an effect of ionic strength on channel gating. Furthermore, we found that BL1249 can activate TREK-2 in high $[K^+_{int}]$ without increasing γ, a result that is consistent with the proposed electrostatic mechanism of action of negatively charged activators.

## Materials and methods

### Molecular biology

For transient transfection in HEK cells, we found that vectors with strong CMV promotors e.g. pFAW were ideal for macroscopic recordings but they made it difficult to obtain single channel recordings. Instead, a modified vector based on the pTK-RL vector (Promega) was used which places TREK-2 under the control of the weaker Thymidine Kinase promotor. The TREK2 construct used here was Isoform b (Isoform 3) with the point mutations M60L and M72L. This removes the alternative translation initiation (ATI) sites and results in the expression of full length TREK-2 only [24, 25]; hereafter, this construct is referred to as TREK2b-FL or 'full length'TREK-2. Additionally, a shortened ATI variant (TREK2bΔ1–72) was used because of its desirable properties for single channel recording (i.e. larger γ); this is referred to as TREK2ΔM$_3$ [24]. Point mutations were introduced by site-directed-mutagenesis and verified by automated sequencing.

### Cell culture and transfection

HEK293 cells were kept in DMEM/F12 culture medium supplemented with 10 v/v% fetal bovine serum. For patch-clamping, cells were seeded into 35 mm dishes covered with 6 mm diameter glass coverslips coated with poly-lysine. Transient transfections were performed using FuGene reagent (Promega). Currents were measured 18–24 h post transfection with either 0.5 ng pFAW channel plasmid, or up to 300 ng pTK channel plasmid for single channel recordings along with 0.25 ng CD8 marker plasmid per culture dish. Coverslips were treated in a 1:5000 dilution of CD8 Dynabeads in PBS for 1–3 min and washed in bath solution before being transferred to the recording chamber. Cells with at least 2 attached beads were considered suitable for recording.

### Electrophysiology

All patch-clamp recordings shown were made in the inside-out configuration using an Axopatch 200B amplifier (Molecular Devices). Pipette solutions contained (in mM): 5 Tris, 2 K2 EDTA, 116 KCl. Internal test solutions contained 5 Tris, 2 K2 EDTA, 116 KCl (120 $K^+$ total) and an additional amount of chloride salt (KCl or NMDG-Cl) to make a total salt concentration of 250, 500 or 1000 mM as indicated. Tris-HCl was chosen over HEPES as a buffer system to minimize errors in $K^+$ concentration incurred by adding large amounts of KOH.

Thick-walled borosilicate capillaries were used to pull microelectrodes. When filled with 120 mM KCl solution they had a resistance of 2–3 MΩ (for macroscopic recordings) or 2–6 MΩ (for single-channel recording depending upon channel density).

Voltage-commands were offset according to the expected Nernst and liquid junction potentials in each test solution, and a range of −100 mV to +100 mV net driving force for $K^+$ was sampled in each condition. Junction potentials were calculated with pClamp which

implements calculations and ionic mobilities described in [26, 27]. As expected for $K^+$ channels in overexpression systems [28], current amplitudes in excised patches varied greatly. To compare IV-curve shapes irrespective of total current, amplitudes were normalized to the value at +70 mV in symmetrical 120 mM KCl solution in Fig 1A.

Macroscopic currents were acquired with 20 kHz sampling rate and low-pass filtered with a 2 kHz Bessel filter (Axopatch 200B onboard filter). Single-channel recordings were acquired at 50 kHz and filtered with 10 kHz.

### Analysis

Single-channel amplitudes were determined with a peak finding algorithm [29] from log-scaled all-point amplitude histograms. All such automatically assigned amplitudes were visually inspected for good fit and corrected by hand if needed. Driving forces were calculated with the Nernst equation assuming a room temperature of 22°C. The open-channel IV-curves in Fig 3b were fit with a third order polynomial using linear regression in R. The model has no mechanistic meaning and was chosen for its ease of implementation, fitting and calculation of the slope (derivative) as a continuous estimate of conductance. Saturation curves of $\gamma$ and internal K concentration (Fig 3c) were fit with a Michaelis-Menten model of the formula:

$$\gamma = \frac{\gamma_{max}[K^+_{int}]}{[K^+_{int}] + K_m}$$

All concentration values in these studies are given as salt concentrations and have not been corrected for activity.

## Results

Our previous work [6, 17] presents several testable predictions about the effect of $[K^+_{int}]$ on flux gated channels. Firstly, based on the 'flux gating' mechanism, increasing the $[K^+_{int}]$ should increase occupancy of the S4 filter $K^+$ binding site. This should then result in channels that are either open, or primed for opening, with filled filter binding sites. Secondly, if BL1249 activates TREK channels primarily via an increase in $K^+$ occupancy of the filter, then its effects on both $\gamma$ and $P_o$ should both be masked in high $[K^+_{int}]$. Such concentration-dependent masking of electrostatic effects on $\gamma$ have been well established in other $K^+$ channels such as the BK and s*haker* channels [30, 31].

### Increased ionic strength reduces TREK2 currents

To test $K^+$-dependent gating, macroscopic TREK-1 and TREK-2 currents were recorded in increased $[K^+_{int}]$. We would expect increased availability of $[K^+_{int}]$ to be followed by increased $P_o$ and a loss of voltage-dependence (i.e. "leak mode") as the flux-gating mechanism becomes overridden by increasing $[K^+_{int}]$ [4, 6]. Also ionic currents are generally expected to scale with the concentration of permeant ion [32].

However, we observed that application of hyperosmolar KCl solutions to inside-out patches was followed by a concentration-dependent reduction in macroscopic TREK-2 currents (Fig 1a). To control for non-specific effects of increased ionic strength, the experiment was repeated using NMDG-Cl of equivalent concentration. This also resulted in a near full reduction of the current over the tested concentration range. It is worth noting that the decrease of outward currents to increasing NMDG-Cl concentration is steeper and monotonous compared to a more graded response in KCl, and this is most prominent in TREK-1 (Fig 1d). Both WT TREK2b-FL and WT TREK1c WT followed this pattern, albeit with variations in the

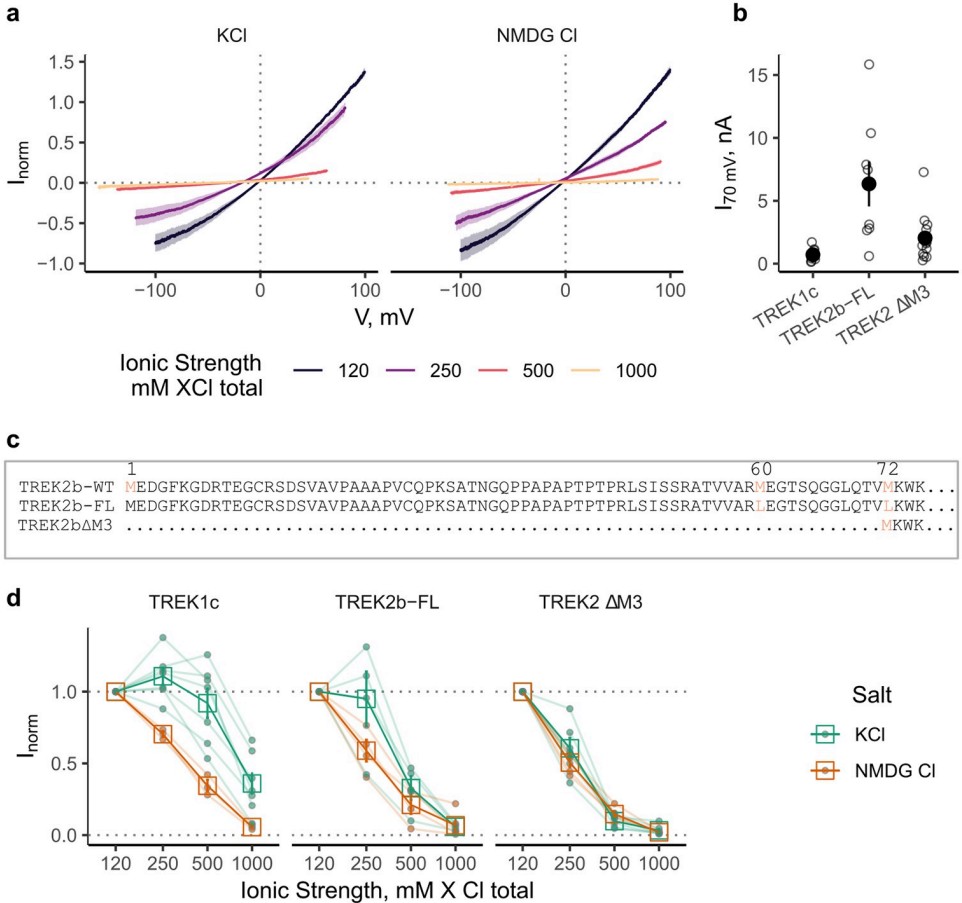

**Fig 1. TREKΔm3 IV-curves in high internal ionic strength. a**) Average normalized IV-curves in varying ionic strength. Shaded area represents SEM, n = 7 and 5 for KCL and NMDG respectively. Note that the IV curves are offset to each other on the x-axis, this is because the voltage-step protocols for each test-solution where modified in order to sample a ± 100 mV driving force range for K⁺. **b**) Absolute current values at 70 mV for the baseline condition (120 mM KCl) for each patch. **c**) Sequence alignment of the N-termini of ATI variants of TREK2 utilized in this study. **d**): Normalized currents in varying ionic strength, corrected for driving force. There was considerable variability in the response between different constructs and between replicates. n > 3. Faint lines and filled circles represent responses of individual patches.

extent of the reduction (Fig 1d). TREK2ΔM3 responded more prominently to an approximate doubling (2.08 fold change) of ionic strength with residual currents of 61±7% for KCl and 51 ±3% in NMDG-Cl (mean and SEM, n = 6 and 5 respectively).

Unfortunately, this unexpected finding prevented a straightforward examination of flux gating by changing K⁺ concentrations. Further experiments were therefore designed to understand this counterintuitive reduction of TREK currents in the presence of high ionic strength.

## Control for non-specific effects of anions and osmotic pressure

Alongside an increase in ionic strength on the intracellular side, the conditions used in the previous experiments introduce an osmotic pressure gradient and therefore may produce a change in membrane tension. This is a possible confounding effect for TREK channels which are also mechanosensitive. Additionally, all experiments were conducted in solutions containing chloride as the only anion with concentrations up to 1M. Therefore, to rule out any

possible effects of these conditions, we measured macroscopic currents in solutions designed to control for these Cl⁻ concentrations as well as imbalanced osmolarity.

TREK2ΔM3 currents were robustly reduced by raising the ionic strength by a factor of $\sim 2$ (250 mM salt concentration; Fig 1). Therefore, we used this channel variant and [K$^+_{int}$] for these further control experiments. Limiting [K$^+_{int}$] to 250 mM also served to avoid and/or reduce effects due to molecular crowding and viscosity of the solution that may occur with very high solute concentrations or ionic strength [30].

Patches in control solution (120 mM KCl) were perfused with either a solution with increased ionic strength and an osmotic gradient across the patch (250 mM KCl as in Fig 1), a solution of increased ionic strength with osmotic gradient across the patch, but fixed Cl⁻ concentration (120 mM KCl, 130 mM KMeSO$_4$), or a solution with an osmotic gradient across the patch, but no increase in ionic strength (120 KCl, 260 mM Mannitol). To avoid any possible irreversible effects on the mechanical properties of the patch due to application of these solutions, patches were discarded after data collection with one test solution. The resulting macroscopic IV-curves and paired current amplitudes are shown in Fig 2.

## Effects of ionic strength are reversible

High ionic strength may weaken salt bridges that are essential for the integrity of protein structures. K$^+$ channels are especially sensitive to [K$^+$]$_{ext}$ due to the filter collapsing into dilated or irreversibly defunct states in low internal K$^+$ [14, 15]. To test for a possible irrecoverable loss of channel function in high ionic strength, application of control (120 mM KCl) and 250 mM KCl solutions was alternated. This resulted in a reversible reduction and recovery of currents. 'Washout' and 'block' completed as fast as the bath perfusion for long periods (>10 min with >10 solution changes, Fig 2d). The reduction in current due to increased ionic strength therefore does not appear to be the result of any irreversible changes in protein structure.

## Single-channel currents reveal saturating conduction

Some of the residual currents after exposure to high ionic strength were sufficiently small enough to discern individual channel openings from the baseline (Fig 3a). Although a formal analysis of $P_o$ was not feasible due to the large and unknown number of channels present in the patches, these traces indicate a strong reduction of $P_o$ with residual short open flickers.

The open-channel IV-curves extracted from these residual currents are shown in Fig 3b. As expected, the unitary current increases with the amount of available K$^+$. This demonstrates that the reduction of macroscopic current seen with increased [K$^+_{int}$] (Fig 1a) is likely due to a reduction of $P_o$. This also serves to explain the shape of the response curve of TREK1c (Fig 1d). Non-permeating NMDG-Cl lowers $P_o$ without raising $\gamma$ resulting in a monotonous decrease, whereas KCl raises $\gamma$ and decreases $P_o$ resulting in a more complex relationship. Furthermore, and as expected for K$^+$ channels with multiple binding sites [33], the relationship of concentration and $\gamma$ is also non-linear and saturating (Fig 3c).

Fitting this to a Michaelis-Menten-model yields values of 343±40 pS (n > 4, patches) maximum conductance with a half maximal concentration of 110±59.8 mM for internal K$^+$ (values for 80 mV, 95% confidence interval of fit estimates). This is in good agreement with similar measurements of the BK channel [34] although care should be taken with interpreting the $K_m$ value here as the estimate lies at the bottom end of the K$^+$ concentrations sampled and more accurate values of such parameters are typically obtained from measurements in symmetrical [K$^+$] and in artificial bilayers [14, 34].

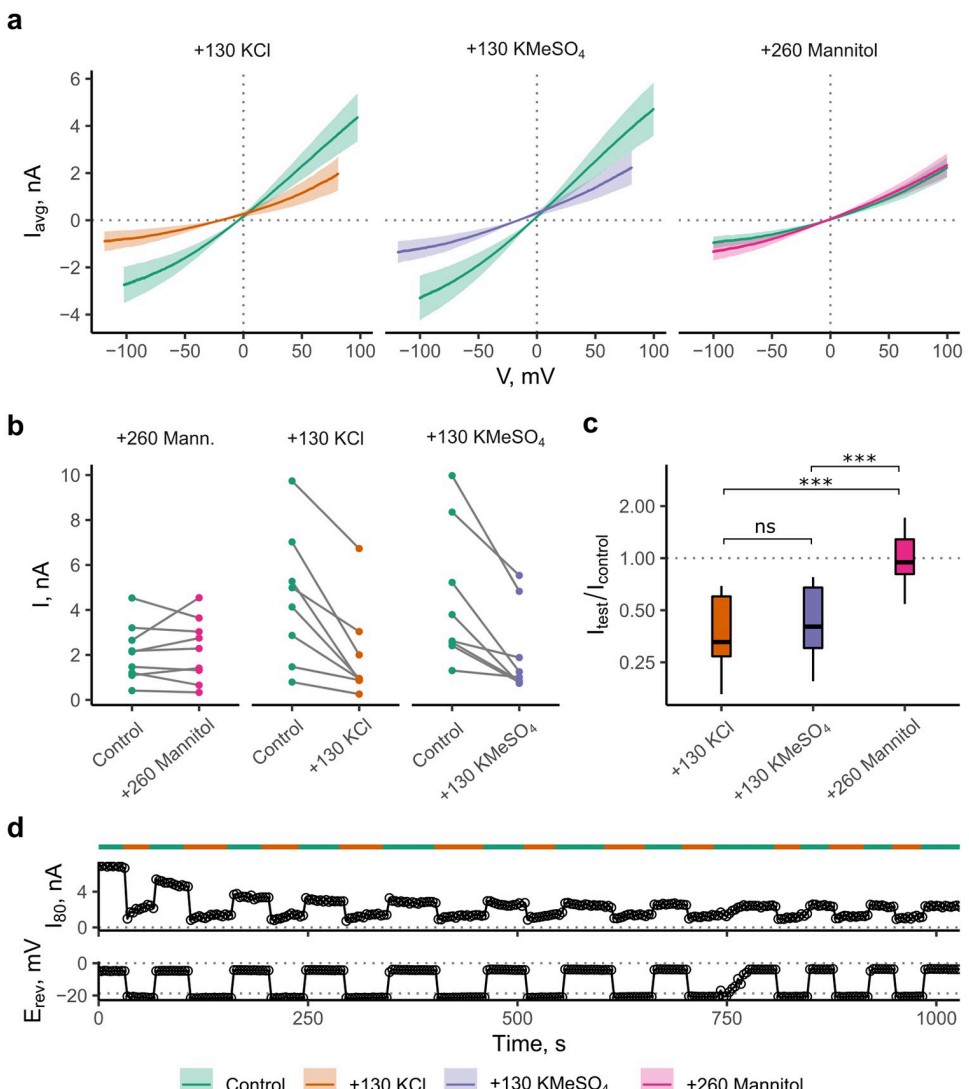

**Fig 2. High ionic strength, not osmotic gradients inhibit TREK2. a**) Mean IV-curves in excised patches. Shaded areas represent ± SEM. One test solution was applied to every patch, then it was discarded (this yields separate control curves for every condition, n >8 paired recordings). **b**) Currents at +95 mV outward driving force. **c**) Relative current change. The mannitol test solution has no appreciable effect on current. Multiple comparisons here were tested with ANOVA and Tukey's post-hoc test. **d**) Time course data of repeated application of 250 mM KCl solution to an inside out patch. Typical example shown from 3 independent measurements. Voltage ramps from -100 to +100 mV were continually applied to an inside-out patch. Full solution exchange was usually achieved in the course of 1 ramp (1.5 s) as shown by the expected change in reversal potential. Channel activity decreased spontaneously over the course of the measurement and then stabilized (rundown). The current size and relative reduction with these test solutions were variable, however every patch tested showed reduced current in response to application of more concentrated salt solutions to the intracellular side (see Fig 2b). By contrast, application of the mannitol solution failed to elicit a consistent reduction of the current. Statistical analysis revealed no differences between the inhibition produced by the two high ionic strength solutions (p = 0.888).

## BL1249 increases $P_o$ but not $\gamma$ in high ionic strength

Previous work suggests an electrostatic mechanism of action of the negatively charged activator, BL1249 (16). Amongst this evidence, single-channel recordings of TREK-1 and TREK-2 demonstrated an increase in apparent $\gamma$ in the presence of BL1249 consistent with electrostatic

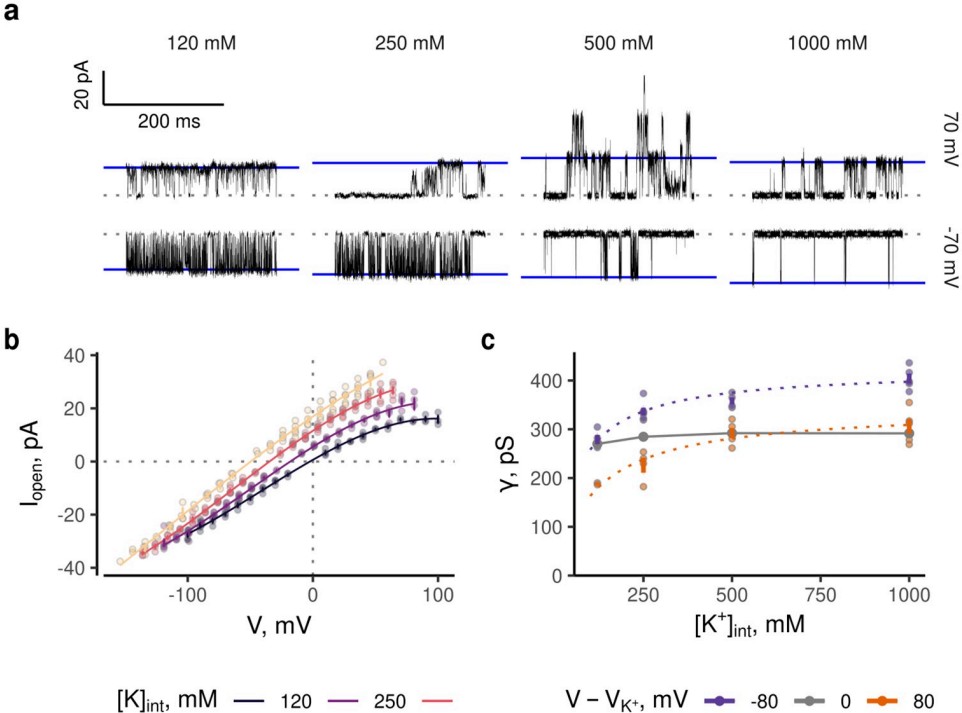

**Fig 3. Single-channel behavior in high internal ionic strength. a)** Example time-courses of low $P_o$ TREK2ΔM3 currents in solutions of varying K$^+$ concentration and ionic strength. Outward (top row) and inward (bottom row) single channel currents at a net driving force of ± 70 mV. Different subconductances and channel flickers are apparent. Fitted amplitudes for the closed (grey) and open state (blue) are shown as horizontal lines, passive leak current was subtracted. **b)** Amplitudes were extracted from short stretches of recordings 0.95 s at voltages ranging from 100 mV either side of the calculated reversal potential in the given K$^+$ gradient. Data collected at high ionic strength (500 mM) represent opening events recorded from residual activity of patches conducting macroscopic currents in the baseline condition (large amount of channels). Lines represent a third order polynomial fit. Number of patches for each reconstructed curve is at least 4 whereas the number of replicates at a given voltage can be less, if the amplitude could not be determined due to lack of openings. **c)** Estimates of $\gamma$ plotted against the internal potassium concentration. Amplitudes were divided by the driving force for K$^+$. The slope at the point of current reversal (y-axis intercept of the fit in panel a) was taken for the 0 mV value.

funneling effects seen with negatively charged amino acids in other K$^+$ channels [14, 15]. We therefore next tested the effect of 3 μM BL1249 (a concentration that increases the apparent conductance, $\gamma$ by 10–20%) on TREK2ΔM3 in a 1M KCl solution. This served to address whether BL1249 can open channels while these electrostatic interactions are weakened and/or whether the closed channels are still responsive to activation; it also examines whether BL1249 has an effect on saturated $\gamma$, i.e. whether it increases $\gamma_{max}$ in the context of the fit in Fig 3c. Furthermore, the proposed electrostatic mechanism of action predicts that BL1249 would left-shift the $\gamma$-concentration curve (i.e. decrease K$_m$) while not affecting $\gamma_{max}$. This is similar to the effect of charged residues in the mouth of the BK channel [14, 15]. Long stretches of a recording containing three channels in 1M KCl are plotted in Fig 4a. As can be seen from the amplitude histograms in Fig 4b, application of BL1249 was followed by a robust increase in $P_o$, but no effect on single channel amplitude; as can be seen from Fig 4d the single-channel IV-curves sampled in these conditions overlapped. All amplitudes before and after application (pairwise, per patch) are plotted in Fig 4c where there was no obvious change conductance (paired t-tests at each voltage with Bonferroni correction, p > 0.05 in all cases).

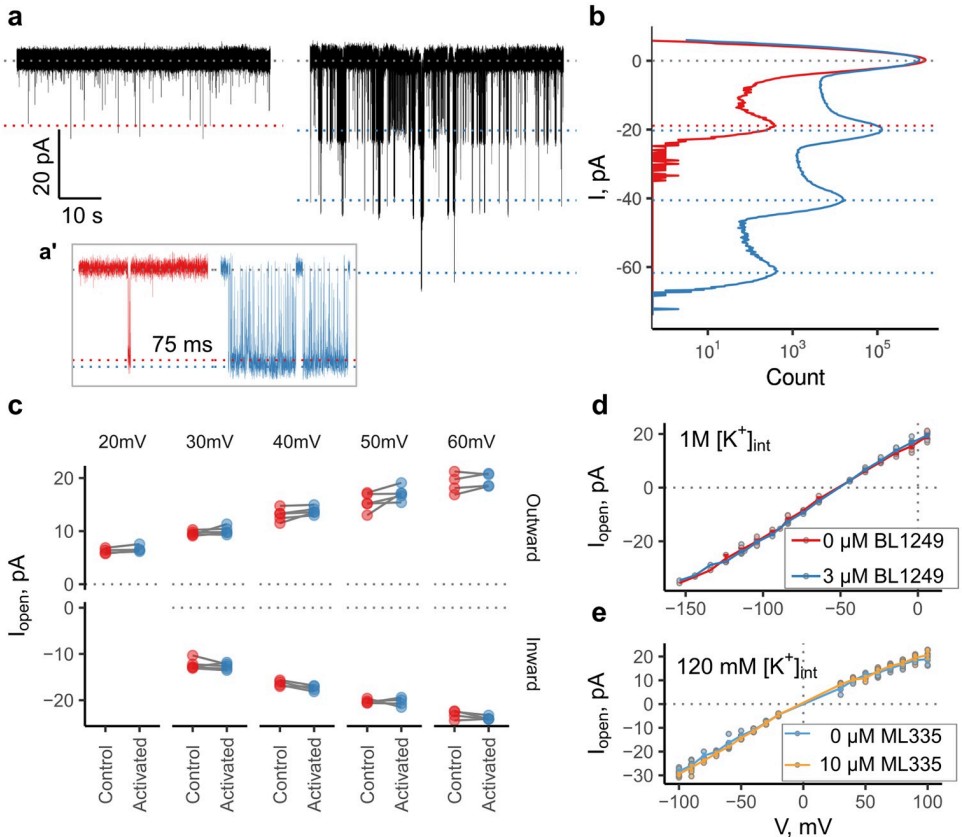

**Fig 4. BL1249 increases *Po* but not *γ* in high ionic strength. a**) Single channel traces of TREK ΔM3 in 1M intracellular KCl at -104 mV (-50 mV driving force). a' Enlarged view of individual openings in each trace (75 ms). **b**) All point amplitude histogram of the recordings shown (partially) in panel a, total length of the recording was 3 min each. **c**) Pair-wise comparison of the single-channel amplitudes plotted. The column label denotes the net driving force in mV. A paired t-test was used to test for a difference in amplitude. There was no statistically significant difference. **d**) Single-channel IV curves of TREK2ΔM3 in 1 M internal KCl concentration, with and without 3μM BL1249. **e**) Single-channel IV curves of TREK2ΔM3 in symmetrical 120 mM K$^+$). No increase in *γ* was observed.

An increase in *γ* is also not a general feature found with all TREK agonists. For example, ML335 is a TREK agonist which binds to a distinct site behind the SF [35], and in agreement with recent evidence [12] we also found that this does not produce any observable increase in *γ* (Fig 4e).

The failure of BL1249 to elicit an increase in *γ* in high concentrations of K$^+$ is consistent with the effect of BL1249 on *γ* being mediated by electrostatic action within the channel pore. However, it also raises the question of how the principal activating effect of BL1249 (via an increase in *P$_o$*) persists in the presence of electrostatic screening, and suggests an additional allosteric effect of drug action on the filter gate (see Discussion).

## Stabilization of the open filter gate conformation

A more detailed view of the open-peak histograms also revealed that the typically skewed inward open-channel current of TREK-2 appears markedly more symmetrical in the presence of both high ionic strength and BL1249 (Fig 5). Thus, in addition to the obvious overall reduction in *P$_o$* via a dominant slow gating process that results in more long closures, high ionic strength also appears to stabilize the open conformation of the filter gate via a fast process by suppressing openings with amplitudes below the mean open level.

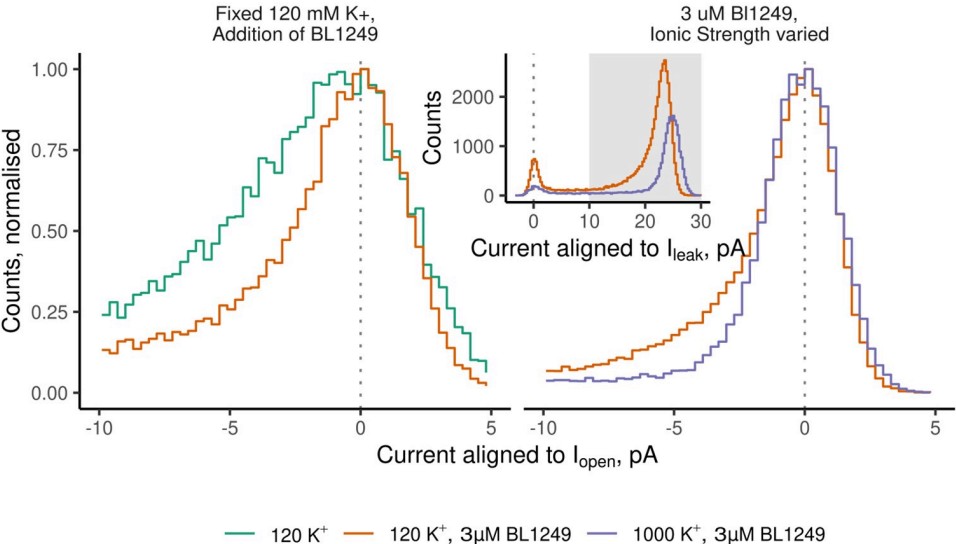

**Fig 5. Combined action of BL1249 and high K⁺ relieves open-state instability.** Comparison of open-state peaks of TREK2ΔM3 in different conditions. The open-state peaks of microscopic TREK2 recordings are often asymmetric or skewed (green trace, left). However, this is substantially reduced in the presence of BL1249 (red trace, left) and also in high ionic strength (purple trace, right). Data in the left panel was taken from a previously published dataset [17]. To compare the shape of the histograms the x-axis was aligned to the peak of the open-channel current, and the y-axis normalized to the peak. See insert for the raw amplitude histogram for the data plotted in the right panel. Membrane voltage was -100 mV except for the trace in 1000 mM K⁺ (-60 mV). At this voltage the open-channel current was closest in amplitude to the matching recording in 120 K⁺.

To determine a comprehensive explanation for the electrostatic effects as well as activation by BL1249, further single channel recordings as well as kinetic analysis are required. While the information gleaned from these recordings is consistent with the data in [17] we would have more confidence with a more complete treatment of the $\gamma$-concentration curve (full range from near 0 to 1M KCl, with and without BL1249) with stringent within-patch comparisons.

Long-term stability of the inside-out patches was negatively affected in strong osmotic gradients. The envisioned experiments also require sampling TREK2 in the full dynamic range of $P_o$. The negligible activity of TREK2 in 1M KCL requires long single-channel recordings to sample openings, but under these conditions the closed states may be difficult or impossible to resolve when the channel is activated by BL1249, especially if more than one channel is present. We also found that TREK2 appears to readily express at high density in the membrane with multi-channel recordings being prevalent. This persists even when preparing transfections with small amounts of TREK2 DNA under the control of a weak promotor.

## Discussion

The results presented demonstrate that TREK channels are inhibited in the presence of high ionic strength and that this represents an intrinsic property of the channels. Unfortunately, this phenomenon therefore precludes experiments in which K⁺ concentrations are systematically varied to examine K⁺-sensitive gating processes within the selectivity filter [6, 12, 17]. However, a more limited experiment designed to probe the influence of electrostatics on the activation by BL1249 suggested that this process is largely independent of electrostatic screening, whereas its effect on $\gamma$ is not and therefore may be due to electrostatic funneling.

The rationale behind using solutions with increased KCl concentrations was to study conditions in which we expect the flux- and K⁺-dependent gating mechanisms of TREK to become

saturated ('leak' modes). However, in these conditions we found the channels to become predominantly closed and not adopt the 'leak' mode (Figs 1 and 3a). While the relationship between occupancy of binding sites and [K+] will be complex, it is nevertheless reasonable to assume that the S4 K+ binding site will become near fully occupied in the conditions tested (up to 1M KCl). Consistent with this, recent crystal structures of TREK-1 in a series of [K+] show that further changes in occupancy and filter distortions do not occur in KCl concentrations above 100 mM [12]. This is also in agreement with measurements of K+ affinity in the filter of prokaryotic K+ channels [36].

K+ channels in general are not functionally impaired in high internal ionic strength and currents have been recorded in comparable K+ gradients for BK, Shaker and Kir2.1 [34, 37, 38]. Instead, high K+ occupancy of these channels brought about by increased [K+] was associated with reduced C-type inactivation and higher channel activity [39] as well as preventing collapse of the pore [16, 40]. Interestingly, in TREK-2 channels, both BL1249 and high ionic-strength resulted in open state peaks in the amplitude histogram that were substantially more symmetric. We have not observed this in other conditions for TREK-2 [23] and it might therefore be explained by a stabilizing effect of BL1249 and K+ on the open state of the filter gate that results in reduced fluctuations of either brief openings, or of subconductance states.

Stable, long closures were observed in high K+ concentrations in which the filter is likely to be stably occupied by K+ (Fig 3a). Therefore it seems likely that the SF can be occupied by K+ and the channel simultaneously rendered non-conductive in addition to K+ depleted closed states of TREK that have emerged from functional [6] and structural studies [12].

The precise molecular mechanisms underlying this inhibitory effect are unknown but may include non-specific effects of ionic strength on the physical properties of membranes which might clearly affect a mechanosensitive channel [41]. Furthermore, although this inhibitory effect of ionic strength is important to be aware of, and affects our ability to dissect the biophysical properties of TREK channels, we do not ascribe any particular physiological significance to this process because it is only observed at supraphysiological ionic concentrations.

Due to their polymodal gating mechanisms, that for some K2P channels includes significant open probability at rest, the whole family are often described as 'leak' channels. However, this terminology is potentially confusing because it erroneously suggests a lack of regulation of gating in these channels. Furthermore, TASK-3 has also been described as a 'GHK leak' channel because in some conditions its steady state IV curve can be reasonably fit with the Goldman current equation [42, 43]. However, K+ channels violate many of the assumptions of the Goldman formalism, most importantly that of independence [44], and this divergence is one of the hallmarks of K+ channel function. K2P channels share many of the features of other tetrameric K+ channels that bring about these properties (e.g. a long asymmetric pore and selectivity filter) [45]. Here we observe such typical saturating behaviors at a single channel level (Fig 3c) [33, 34, 46] and non-linear single-channel IV curves in symmetrical K+ concentrations (Fig 3b). These properties are not predicted by the Goldman equation and are fully expected for K+ channels with complex permeation mechanisms in which permeating ions interact with the pore. While this result is hardly surprising given our current knowledge of K2P channel structures, it remains, to the best of our knowledge, the first direct observation of multi-ion permeation in K2P channels from electrophysiological recordings. The satisfactory fit of macroscopic IV curves with the Goldman equation is entirely coincidental due to the shape of the channel activation curve (i.e. the $P_o$-voltage relationship) and the single-channel IV curve. It cannot predict that these channels are 'open rectifiers' or 'leak' channels. Thus although the usefulness of the Goldman equation in describing electrical membrane phenomena remains unmatched [47], its application to the mechanisms of K+ channel permeation is less well justified.

In our previous work studying the mechanism of action of BL1249 it was proposed that increased occupancy of $K^+$ in the pore may lead to stabilization of the filter and therefore activation, and this principle was assumed to be applicable to all $K^+$ channels gated at the filter [17]. In agreement with this electrostatic mechanism, the increase in $\gamma$ accompanying BL1249 activation does not occur in the presence of high KCl concentrations, but channel open probability is still increased by BL1249 under such conditions, indicating that the activatory effect of the drug is less sensitive to changes in the electrostatic environment. It also suggests that BL1249 may have an allosteric stabilizing effect on the filter gate in TREK channels possibly similar to the way in which certain antagonists affect the filter gate mechanism [23]. Ultimately, this needs to be addressed with genuine single channel recordings of wild-type channels in preparations that allow high resolution measurements for kinetic analysis. However, the findings presented here, and our previous study [23], indicate the many challenges required to make such measurements and determine the direct concentration-dependent effects of the permeating ion especially when they become masked by the inhibitory effects of increased ionic strength.

## Supporting information

**S1 File.**
(ZIP)

## Acknowledgments

We thank members of the Tucker group for helpful discussions.

## Author Contributions

**Conceptualization:** Linus J. Conrad, Stephen J. Tucker.

**Data curation:** Linus J. Conrad.

**Formal analysis:** Linus J. Conrad, Peter Proks, Stephen J. Tucker.

**Funding acquisition:** Stephen J. Tucker.

**Investigation:** Linus J. Conrad, Peter Proks.

**Methodology:** Linus J. Conrad, Peter Proks.

**Project administration:** Stephen J. Tucker.

**Software:** Linus J. Conrad.

**Supervision:** Stephen J. Tucker.

**Validation:** Linus J. Conrad, Peter Proks.

**Visualization:** Linus J. Conrad, Peter Proks.

**Writing – original draft:** Linus J. Conrad, Stephen J. Tucker.

**Writing – review & editing:** Peter Proks, Stephen J. Tucker.

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
