## [Decision Letter · Decision Letter 0]

5 Aug 2021

PONE-D-21-22346

Effects of Ionic Strength on Gating and Permeation of TREK-2 K2P channels

PLOS ONE

Dear Dr. Tucker,

Thank you for submitting your manuscript to PLOS ONE. After careful consideration, we feel that it has merit but does not fully meet PLOS ONE’s publication criteria as it currently stands. Therefore, we invite you to submit a revised version of the manuscript that addresses the points raised during the review process.

Both reviewers found your work interesting and would be important for ion channel physiology people. While reviewer one does not have any specific comment, reviewer two and I have few comments. You can find all observations at the end of this message and I hope you can answer them without trouble.  

We look forward to receiving your revised manuscript.

Kind regards,

Jorge Arreola, PhD

Academic Editor

PLOS ONE

Journal Requirements:

"None"

"This work was supported by the Wellcome Trust and the Biotechnology and Biological Sciences Research Council (BBSRC)."

"This work was supported by the Wellcome Trust and the Biotechnology and Biological Sciences Research Council (BBSRC). The funders had no role in study design, data collection and analysis, decision to publish, or preparation of the manuscript."

Additional Editor Comments:

Based on my evaluation, I concur with the reviewer´s comments. However, I have three criticisms/suggestions that the I hope will help the authors to improve their work.

The unexpected finding that high concentrations of intracellular K decrease TREK K2P channel open probability by Tucker´s group is quite interesting. The authors propose that high ionic strength plays a role in this phenomenon; however, they also suggest a stably occupied filter by K underlies the reduced open probability caused by high K concentration. Thus, the authors need to make their conclusion more explicit to understand the role of ionic strength and selectivity filter occupation.  The data presented in Figure 4b shows that the conductance increased with the K concentration. However, I found the analysis less credible. The Michaelis-Menten equation used to analyse the conductance vs K concentration predicts that the conductance is 0 at 0 K. This does not seem to be the case. Also, the estimated Km (110 mM) is outside the K concentration range used in the experiments, making this Km value less realistic. Finally, what is the explanation for the voltage-dependent effect of K on the conductance vs K concentration curves?   I suggest combining Figures 3 and 4 because they are closely related.      

Reviewers' comments:

Reviewer's Responses to Questions

**Comments to the Author**

1. Is the manuscript technically sound, and do the data support the conclusions?

Reviewer #1: Yes

Reviewer #2: Yes

2. Has the statistical analysis been performed appropriately and rigorously? 

Reviewer #1: Yes

Reviewer #2: Yes

3. Have the authors made all data underlying the findings in their manuscript fully available?

Reviewer #1: Yes

Reviewer #2: Yes

4. Is the manuscript presented in an intelligible fashion and written in standard English?

Reviewer #1: Yes

Reviewer #2: Yes

5. Review Comments to the Author

Reviewer #1: This is a very convincing manuscript showing that TREK-2 is inhibited by the ionic strength of the internal medium. The control experiments are adequate. The authors also show that the effect of the opener BL149 on channel open probability is retained at high ionic strength but not the effect on channel unit conductance. This provides insight into how BL1249 modulates TREK-2. The data are well presented and discussed. The manuscript is very well written. We agree with the authors that the term "leak" for these channels is inappropriate when considering their electrophysiological properties, and that it would be better to return to the original description of background channels.

Lines 255-256: please replace Fig. 1C by Fig. 1D

No other specific comments

Reviewer #2: In this study, Conrad et al investigating for the K+ dependent gating process of TREK-2 channel by using high concentrations of internal K+, found an inhibitory effect induced by the high K+ concentration of the recording solution. The authors present compelling evidence that this behavior is indeed dependent on the high ionic-strength of the internal solution. The authors also showed that the activator BL1249 was still able to increase the Po in high ionic-strength. I found this to be an interesting paper and observation that is well done, and I feel this will be of interest to the ion channel field. I have a few comments:

1. How to reconcile data from high ionic-strength and the activator BL1249? In discussion is proposed: “a poorly conductive but K+-filled filter conformation may also exist in TREK channels, and it is this conformation that presumably underlies the reduced Po in very high K+ concentrations.” However, BL1249 was still able to increase Po, any idea of the mechanism?

2. There are some figures not properly indicated in the text, e.g., page 8, line 145 Fig. 3a should say Fig. 4a. Page 8, line 148, Fig. 3b should say 4b. Page 9, line 175, (See Fig. 5.1c) should say (see Fig. 1d) and Fig 1c should say Fig. 1d. Page 15, line 288, Fig. 3b should say 4b.

6. PLOS authors have the option to publish the peer review history of their article (what does this mean?). If published, this will include your full peer review and any attached files.

Reviewer #1: No

Reviewer #2: No

---

## [Author Response · Author response to Decision Letter 0]

15 Sep 2021

Please see the attached file which contains a detailed response to the editor and reviewer's comments.

---

## [Decision Letter · Decision Letter 1]

23 Sep 2021

Effects of ionic strength on gating and permeation of TREK-2 K2P channels

PONE-D-21-22346R1

Dear Dr. Tucker,

We’re pleased to inform you that your manuscript has been judged scientifically suitable for publication and will be formally accepted for publication once it meets all outstanding technical requirements.

Kind regards,

Jorge Arreola, PhD

Academic Editor

PLOS ONE

Additional Editor Comments (optional):

Thank you for clarifying my comments.

Reviewers' comments:

Reviewer's Responses to Questions

**Comments to the Author**

1. If the authors have adequately addressed your comments raised in a previous round of review and you feel that this manuscript is now acceptable for publication, you may indicate that here to bypass the “Comments to the Author” section, enter your conflict of interest statement in the “Confidential to Editor” section, and submit your "Accept" recommendation.

Reviewer #1: All comments have been addressed

Reviewer #2: All comments have been addressed

2. Is the manuscript technically sound, and do the data support the conclusions?

Reviewer #1: Yes

Reviewer #2: (No Response)

3. Has the statistical analysis been performed appropriately and rigorously? 

Reviewer #1: Yes

Reviewer #2: (No Response)

4. Have the authors made all data underlying the findings in their manuscript fully available?

Reviewer #1: Yes

Reviewer #2: (No Response)

5. Is the manuscript presented in an intelligible fashion and written in standard English?

Reviewer #1: Yes

Reviewer #2: (No Response)

6. Review Comments to the Author

Reviewer #1: (No Response)

Reviewer #2: (No Response)

7. PLOS authors have the option to publish the peer review history of their article (what does this mean?). If published, this will include your full peer review and any attached files.

Reviewer #1: No

Reviewer #2: No

---

## [Editor Report · Acceptance letter]

29 Sep 2021

PONE-D-21-22346R1 

Effects of ionic strength on gating and permeation of TREK-2 K2P channels 

Dear Dr. Tucker:

I'm pleased to inform you that your manuscript has been deemed suitable for publication in PLOS ONE. Congratulations! Your manuscript is now with our production department. 

Kind regards, 

on behalf of

Dr. Jorge Arreola 

Academic Editor

PLOS ONE